# Prevalence and determinants of contraceptive use among men in Tanzania: Analysis of the 2022 demographic and health survey

Pankras Luoga[1]*, Jovinary Adam[2], Siri A. Abihudi[3]

1 Department of Development Studies, Muhimbili University of Health and Allied Sciences, Dar es Salaam, Tanzania, 2 An independent researcher working in Tabata, Dar es Salaam, Tanzania, 3 Institute of Traditional Medicine, Muhimbili University of Health and Allied Sciences, Dar es Salaam, Tanzania

* luoga.pankras1@gmail.com

## Abstract

Globally, contraceptive use is an important strategy in fighting maternal and neonatal deaths. The spacing and avoiding unplanned pregnancies while providing woman with enough time recovering her health and newborns growing. However, the contraception uses in developing countries including Tanzania is low and is worse among men, culturally regarded as the dominant decision makers in sexual relationships. This study intended to assess prevalence and determinants of the contraceptive use among Tanzanian men using the Tanzania Demographic and Health Survey (TDHS) 2022. The study analyzed secondary data collected using cross-sectional study design of weighted 5763 men obtained from the TDHS 2022. A dependent variable was contraceptive use and independent variables were man's demographic and socio-economic characteristics. Bivariate and multivariable analysis were conducted and p-value<0.05 determined a significant factor. The prevalence of contraceptive use among Tanzanian men is 26%. The logistic regression showed men aged 45–49 years (aOR=3.08, 95% CI = 1.90-5.01) had higher odds to use contraceptive compared to men aged 15–19. Men with higher education (aOR=2.94, 95% CI = 1.79-4.84) had higher odds to use contraceptive compared to those with informal education, from rich quantile (aOR=1.42, 95% CI = 0.92-1.46) had higher odds compared to poor. Men with five and above children (aOR=1.62, 95% CI = 1.08-2.43) had higher odds to use contraceptive compared to those with no child. Those desired no more child had odds of 1.4 times higher to use contraceptive (aOR=1.40, 95% CI = 1.05-1.88), men who heard family planning on radio (aOR=1.39, 95% CI = 1.16-1.66) had higher odds of using contraceptive to those who did not. The contraceptive use among Tanzanian men is generally low with a prevalence rate of only 26% and was associated with man's age, education level, wealth index, number of children, and occupation. More tailored programs targeting men to increase their education level particularly health education are crucial in increasing men's contraceptive use in Tanzania.

**Data availability statement:** The data used in the analysis are available online and can be requested from the DHS custodian website https://www.dhsprogram.com/data/data-set_admin. Users need to register to access the dataset.

**Funding:** The authors received no specific funding for this work.

**Competing interests:** The authors have declared that no competing interests exist.

## Introduction

### Background

The economic prosperity, well-being of individuals, and environmental sustainability of a nation are contingent upon its capacity to effectively manage population growth [1]. Various characteristics such as age, gender, education level, number of children a person have and wealth influence the utilization of contraceptive methods [2–5]. This has led to the promotion of diverse contraceptive methods in sub-Saharan Africa, where individuals used to have a high number of children in the past, regardless of their gender [1,6]. Recently, there has been a rise in the variety of contraceptive methods available in Tanzania, and the desire to use the methods differs across different age groups, sex and geographical location [4,7–9]. The younger population has been found to be hesitant in utilizing contraceptive methods in comparison to the elderly. There is a correlation between favorable wealth conditions and a person's high level of education with an increased likelihood of utilizing contraception, as opposed to individuals who are impoverished and less educated [10,11]. The more the number of children a person has, the greater the likelihood that they will use a contraceptive method in both genders, in comparison to those who have none [4]. Regarding gender, research indicates that the utilization of contraception among males remains relatively low in comparison to females [3,12].

Furthermore, it is evident that having access to information on family planning has significant impact on the reproductive outcome [13]. Various individuals acquire knowledge about contraceptive from different sources such as media and hospitals [4,14,15]. However, research has revealed that men, in particular, face restricted access to certain communication channels, notably hospitals [14].

In recent times, the society has witnessed an increase in types of family planning methods in Tanzania. However, men are reported to have more knowledge and use two types of contraception, condoms and injectable [16,17]. Previous studies reported that the use of family planning without physicians' consultations may have undesirable negative effects on the users [18]. The use and demand for FP seem to vary among ages, religion and area of residence (urban/rural) [3,18]. Tanzania Demographic and Health Survey (TDHS) reported that 38% of currently married women are using any contraceptive method, including 31% who are using any modern method and 7% of women using any traditional method. In addition, among sexually active, unmarried women, 45% use any contraceptive method, including 36% using any modern method and 8% using any traditional method [16]. However, there is limited empirical studies regarding the prevalence and factors influencing the use of contraceptive among men in the Tanzanian context. These factors can be attributed to several variables that will be revealed in the course of this study. Therefore, this study aimed to assess the prevalence and the factors associated with contraceptive uses among Tanzanian men using TDHS 2022.

## Methods and tools

This secondary study used data obtained from a cross-sectional study survey of 8th Tanzania Demographic and Health Survey of 2022. In Tanzania, the program is

implemented by the National Bureau of Statistics (NBS) with the financial support from the United States Agency for International Development (USAID).. The sample design for the 2022 TDHS-MIS was carried out in two stages. The first stage involved selection of sampling points (clusters) consisting of enumeration areas (EAs) delineated for the 2012 Tanzania Population and Housing Census [19]. A total of 629 clusters were selected. Among the 629 EAs, 211 were from urban areas and 418 were from rural areas. In the second stage, 26 households were selected systematically from each cluster [16].

Interviews were conducted with all women and men between the ages of 15 and 49 who were usual residents or visitors slept in the selected households the night preceding the day of the survey. No additional ethical approval was required beside the ethical considerations followed by DHS surveys. Ethical approval was obtained from participants prior to data collection by DHS program. The confidentiality and privacy of respondents are rigorously upheld throughout the DHS survey. Detailed description of the methodology and questionnaires used in the survey is available in the final report of the TDHS 2022 [16].

Concerning this study; the MEASURE DHS approved the use of the datasets after reviewing our concept note that was submitted to them. The datasets are available to the public for free at the DHS Program (https://dhsprogram.com/data/new-user-registration.cfm).

The analysis used the men's file (MR) to elicit men's information towards the use of contraceptive in Tanzania. In addition, this analysis involved some variables that were used in other similar studies conducted previously [8,19].

### Variables

**Independent variables** involved men's demographic characteristics including age, education level, marital status (not in union and in union), area of residence, wealth quintile, occupation, parity, desire for another child, knowledge on contraceptive methods, heard family planning on radio, contraceptive is women's business, women using contraception become promiscuous, watching television, discussion with health worker about contraception [8]. As indicated in (Fig 1)

**Dependent variable** was the use of the Family Planning method which was dichotomized into yes or no responses. In the analysis, all the variables with no data were treated as missing.

**Statistical analysis;** data were analyzed using Stata version 17 software with application of svy. Command to account for complex sampling design and non-response rate. This provided nationally representative estimates as per by the DHS recommendations [20]. Univariate, binary with chi-square and multivariable analysis models were used to find the association and the odds of using contraceptive based on the socio-demographic characteristics entered in the models. The chi-square was used to determine the association of the variables. Then variables which showed to be significant at p-value<0.25 in the model were tested for potential multi-collinearity to ensure the independent variables do not relate to each other except to dependent variable. Those passed the test were entered in the multivariable logistic regression analysis model to be able to control for other variables. The threshold value of p-value <0.05 was used to determine significance of the variable.

### Ethical statement

The study analyzed the collected data from Demographic Health Survey (DHS) which had already obtained ethical clearance from Tanzanian National Bureau of Statistics (NBS) for data collection hence this study did not need another ethical clearance. However, permission to use the data was requested from the DHS custodian USAID MEASURES.

### Results

### Background characteristics of study participants

The study involved 5763 male respondents, with a mean age of 29 years. Most respondents were aged 15–19 years (25.1%), followed by 20–24 years (16.2%). Majority had primary education (54.4%), with a minority having higher

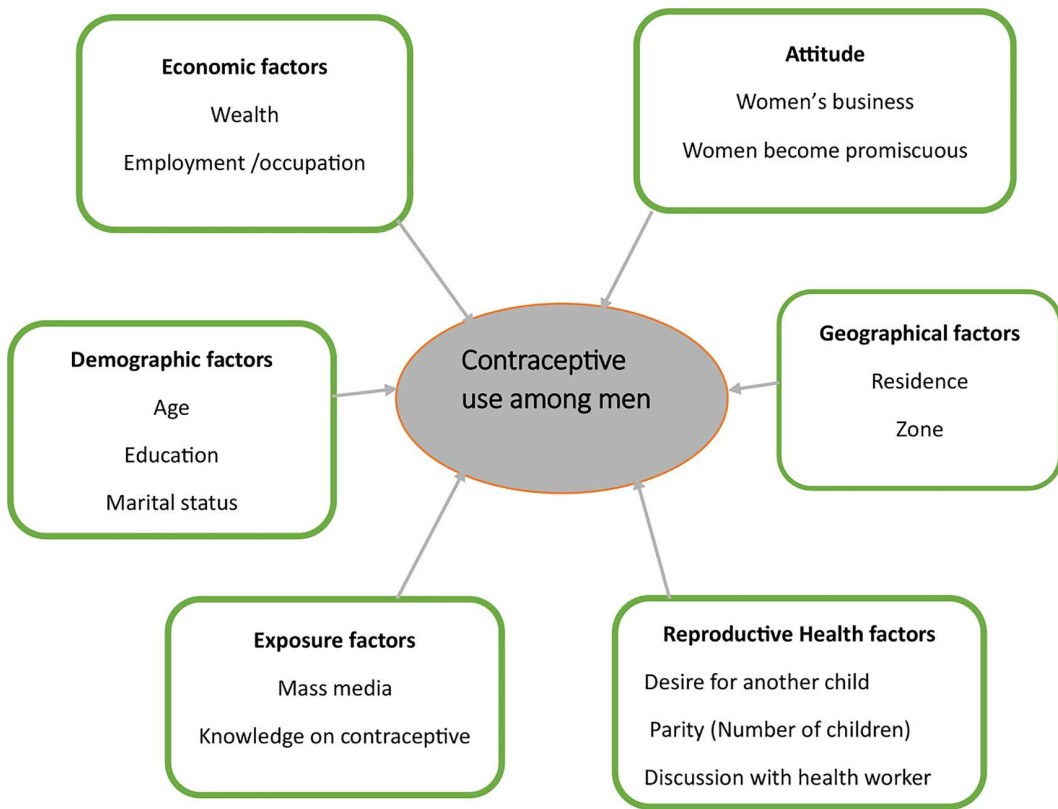

**Fig 1. The conceptual framework of the use of contraceptive use among men.**

education (3.4%). More than a half (66.4%) residing in rural areas. More than a half (51.0%) were living in union, poorest (33.3%) richest (46.0%). More than two thirds (82.3%) had knowledge on any contraceptive method, while half (52%) listened to a radio at least once a week (Table 1).

## Prevalence of contraceptive use among men across geographical zones

The overall prevalence of contraceptive use among men aged 15–49 in Tanzania is 26% (95% CI:24.02, 27.28) whereby most of them (90.6%) use modern contraceptive methods, and the rest 9.4% use traditional contraceptive methods (Fig 2). The prevalence varied significantly across zones. Modern contraceptive use is higher (38%) and (32%) in the Southern Highlands and South West Highlands respectively and lowest (8%) in Zanzibar. Across zones, traditional contraceptive use ranges from 0.2% to 4% in Southern and South West Highlands respectively (Table 2).

## Determinants of contraceptive use among men

The results as shown in (Table 3) from logistic regression analysis revealed that the odds of that men aged 20–24 years (aOR=2.97, 95% CI = 2.19-4.04); 25–29 years (aOR=2.96, 95% CI = 2.11-4.15); 30–34 years (aOR=3.11, 2.11-4.61); 35–39 years (aOR=3.12, 95% CI = 2.07-4.71); 40–44 years (aOR=2.89, 95% CI = 1.84-4.53) and 45–49 years (aOR=3.08, 95% CI = 1.90-5.01) had higher odds to use contraceptive compared to men aged 15–19. Men with primary education (aOR=1.88, 95% CI = 1.41-2.51); secondary education (aOR=2.04, 95% CI = 1.46-2.84) and higher education (aOR=2.94, 95% CI = 1.79-4.84) had higher odds of using contraceptive compared to those with informal education. The odds of

**Table 1. Socio-demographic characteristics and association with contraceptive use among men of reproductive age (15–49 years) in Tanzania (using TDHS 2022).**

| Variables | | Frequency | Percentage | Ever used contraceptive (%) | | |
| --- | --- | --- | --- | --- | --- | --- |
| | | | | No | Yes | P-value |
| Age | 15–19 | 1444 | 25.1 | 90.3 | 9.7 | <0.001 |
| | 20-24 | 934 | 16.2 | 69.7 | 30.3 | |
| | 25-29 | 850 | 14.8 | 69.0 | 31.0 | |
| | 30-34 | 765 | 13.3 | 67.6 | 32.4 | |
| | 35-39 | 693 | 12.0 | 68.7 | 31.3 | |
| | 40-44 | 607 | 10.5 | 71.4 | 28.6 | |
| | 45-49 | 469 | 8.1 | 68.0 | 32.0 | |
| Residence | Urban | 1938 | 33.6 | 71.0 | 29.0 | 0.0053 |
| | Rural | 3825 | 66.4 | 76.1 | 23.9 | |
| Education | None | 574 | 10.0 | 87.0 | 13.0 | <0.001 |
| | Primary | 3134 | 54.4 | 73.7 | 26.2 | |
| | Secondary | 1858 | 32.2 | 73.5 | 26.5 | |
| | Higher | 197 | 3.4 | 55.9 | 44.1 | |
| Wealth | Poor | 1920 | 33.3 | 80.5 | 19.5 | <0.001 |
| | Middle | 1191 | 20.7 | 74.6 | 25.4 | |
| | Rich | 2652 | 46.0 | 69.9 | 30.1 | |
| Occupation | Unemployed | 871 | 15.1 | 90.8 | 9.2 | <0.001 |
| | Employed | 4892 | 84.9 | 71.5 | 28.5 | |
| Marital status | Not in union | 2826 | 49.0 | 77.5 | 22.5 | **<**0.001 |
| | In union | 2937 | 51.0 | 71.4 | 28.6 | |
| Parity | 0 | 2555 | 44.3 | 81.3 | 18.7 | <0.001 |
| | 1-2 | 1259 | 21.9 | 67.4 | 32.6 | |
| | 3-4 | 1038 | 18.0 | 69.7 | 30.3 | |
| | 5 and above | 911 | 15.8 | 70.1 | 29.9 | |
| Desire for another child | Have another | 2213 | 38.4 | 72.8 | 27.2 | <0.001 |
| | Undecided | 265 | 4.6 | 71.1 | 28.9 | |
| | Wants no more | 416 | 7.2 | 64.4 | 35.6 | |
| | Sterilized | 2869 | 49.8 | 77.4 | 22.6 | |
| Knowledge on contraceptive methods | No | 1022 | 17.7 | 90.30 | 9.70 | <0.001 |
| | Yes | 4741 | 82.3 | 70.97 | 29.03 | |
| Heard family planning on radio | No | 2062 | 35.8 | 82.71 | 17.2 | <0.001 |
| | yes | 3701 | 64.2 | 69.76 | 30.24 | |
| Contraception is women's business | Disagree | 2653 | 46.0 | 73.13 | 26.87 | <0.001 |
| | Agree | 2520 | 89.8 | 71.93 | 28.07 | |
| | Don't know | 590 | 10.2 | 90.64 | 9.36 | |
| Women using contraception become promiscuous | Disagree | 2317 | 40.2 | 72.14 | 27.86 | <0.001 |
| | Agree | 2881 | 50.0 | 72.47 | 27.53 | |
| | Don't know | 565 | 9.8 | 93.45 | 6.55 | |
| Frequency of listening radio | Not at all | 1193 | 20.7 | 80.4 | 19.6 | <0.001 |
| | Less than once a week | 1575 | 27.3 | 73.6 | 26.4 | |
| | At least once a week | 2994 | 52.0 | 72.4 | 27.6 | |
| Frequency watching television | Not at all | 1326 | 23.0 | 81.1 | 18.9 | <0.001 |
| | Less than once a week | 1709 | 29.7 | 73.7 | 26.3 | |
| | At least once a week | 2728 | 47.3 | 71.6 | 28.4 | |

*(Continued)*

**Table 1.** (Continued)

| Variables | | Frequency | Percentage | Ever used contraceptive (%) | | |
|---|---|---|---|---|---|---|
| | | | | No | Yes | P-value |
| Discussion with health worker about contraception | No | 5076 | 88.1 | 75.22 | 24.78 | 0.0011 |
| | Yes | 687 | 11.9 | 68.28 | 31.72 | |

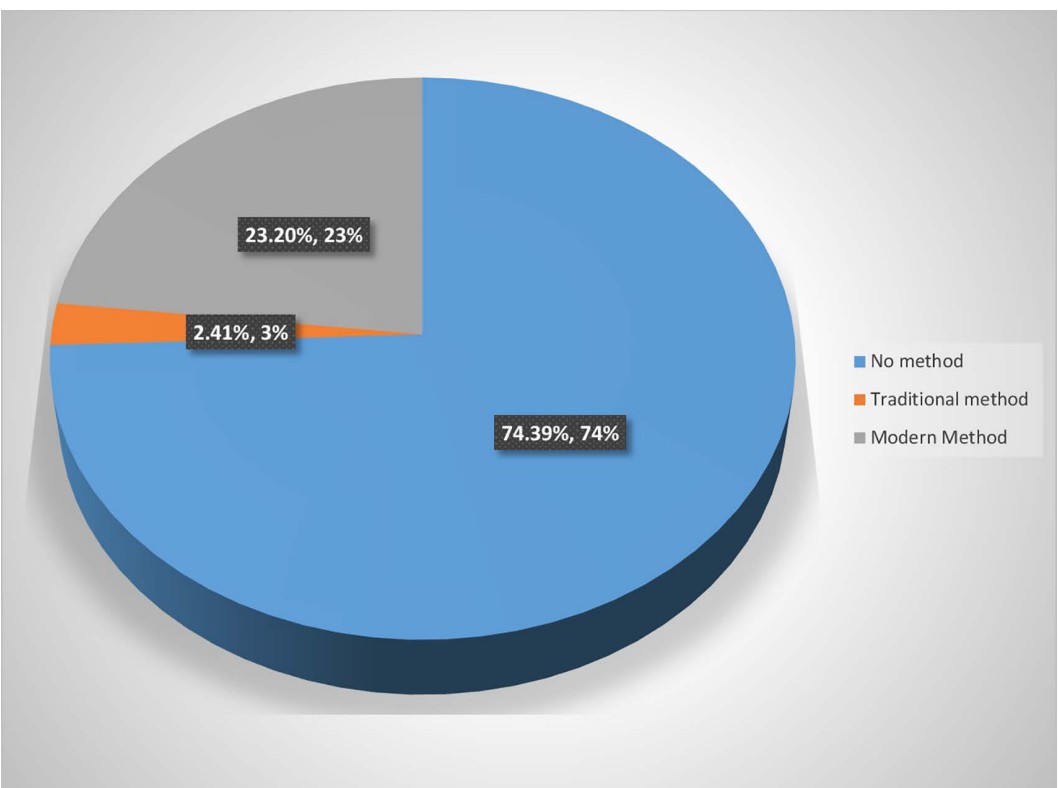

**Fig 2. Prevalence of contraceptive use among the methods.**

modern contraception use were significantly high among men from middle wealth index (aOR=1.23, 95% CI = 0.98-1.56) and rich (aOR=1.42, 95% CI = 0.92-1.46) compared to poor. Employed men (aOR=2.01, 95% CI = 1.34-3.02) had higher odds to use contraceptive compared to unemployed men. Compared with men with no child, men with one to two (aOR=1.51, 95% CI = 1.12-2.02) three to four (aOR=1.53, 95% CI = 1.05-2.24) and five and above (aOR=1.62, 95% CI = 1.08-2.43) children had higher odds to use contraceptives. The men who desired no more child had odds of 1.4 times higher to use contraceptive (aOR=1.40, 95% CI = 1.05-1.88), men with knowledge on contraceptive methods (aOR=1.95, 95% CI = 1.45-2.62) as well as men who heard family planning in the radio (aOR=1.39, 95% CI = 1.16-1.66). Moreover, men who didn't perceive that a woman using contraception becomes promiscuous (aOR=0.42, 95% CI = 0.27-0.66) had lower odds to use any of the contraceptive methods compared to their counterparts.

## Discussion

The study assessed the prevalence and determinants of contraceptive use among Tanzanian men. The overall prevalence of contraceptive use among men aged 15–49 in Tanzania stands at 26%, which is very low compared target of reaching

Table 2. Prevalence of the contraceptives (Modern and tradition) use among men 15-49 years across zones (Using TDHS 2022).

| ZONE | NO METHOD | | | TRADITIONAL METHOD | | | MODERN METHOD | | |
|---|---|---|---|---|---|---|---|---|---|
| | N | % | 95% CI | N | % | 95% CI | N | % | 95% CI |
| Overall prevalence 26% (95% CI:24.02, 27.28) | 4,287 | 74.4 | 72.7-75.9 | 139 | 2.4 | 1.9-2.9 | 1,337 | 23.2 | 21.7-24.8 |
| Western | 419 | 83.6 | 79.78 - 86.78 | 9 | 1.7 | 0.72 – 4.07 | 74 | 14.7 | 11.58 -18.49 |
| Northern | 456 | 72.4 | 66.67 - 77.39 | 18 | 2.8 | 1.67- 4.75 | 157 | 24.8 | 20.00 -30.38 |
| Central | 417 | 72.2 | 67.77 - 76.21 | 15 | 2.6 | 1.34 – 5.08 | 145 | 25.2 | 21.46 – 29.32 |
| Southern Highlands | 225 | 59.9 | 54.39 - 65.18 | 8 | 2.2 | 1.22 – 3.77 | 143 | 37.9 | 32.75 – 43.43 |
| Southern | 222 | 76.3 | 70.18 - 81.56 | 1 | 0.2 | 0.20 – 1.14 | 68 | 23.5 | 18.32 – 29.63 |
| South West Highlands | 338 | 64.3 | 59.43 - 89.40 | 22 | 4.1 | 2.78 – 6.06 | 166 | 31.6 | 27.18 – 36.28 |
| Lake | 1,349 | 79.6 | 75.78 - 83.00 | 32 | 1.9 | 1.19 – 3.09 | 312 | 18.5 | 15.26 – 22.13 |
| Eastern | 691 | 70.8 | 66.91 - 74.31 | 28 | 3.0 | 1.74 – 4.96 | 257 | 26.3 | 22.91 – 30.00 |
| Zanzibar | 170 | 89.0 | 86.22- 91.23 | 6 | 3.0 | 2.04 - 4.48 | 15 | 8.0 | 5.88 – 10.78 |

40% to Tanzania National Family Planning Costed Implementation Plan (NFPCIP) 2019–2023 [21]. Despite policy commitments under Tanzania's NFPCIP and advocacy strategies promoting male involvement in family planning, the 26% prevalence rate of contraceptive use among men aged 15–49 remains markedly low. This figure not only lags behind the national strategic vision but also fails to meet regional and global benchmarks for equitable reproductive health engagement.

In summary, the significant factors to contraceptive use included age, education level, wealth quantile, occupation, parity and knowledge on contraceptive use. Prior descriptive statistics showed that the study population was mostly composed of young generation mostly at reproductive age as it indicated by the average age of 29 years. Majority of population found in rural area (66%), and this showed the need to focus interventions on addressing the contraceptive use challenges encountered by this group on including both those men living in union (51%) and those not living in a union (49%). The substantial proportion (80%) of the population possessed knowledge of contraception methods indicated a promising likelihood of informed decision-making and adoption of family planning methods as reported in other studies [2,17].

This emphasized the importance of education and awareness in promoting contraceptive use and responsible reproductive informed choices [14]. With high percentage (82%) of population having knowledge on contraceptive methods shows a high chance of the population making the right decision and start using family planning methods thus reducing the population [22]. Moreover, radios (52%) have demonstrated effectiveness as a platform for disseminating family planning messages in this study and other studies [4,14,23]. Hence, they may also prove beneficial in advocating for other health-related issues within communities.

Among those who use contraceptive, majority (90%) preferred modern methods including condoms in comparison with traditional methods like withdrawal. However, modern methods are reported to be more effective, reliable and increased uptake as it is supported by a study done in sub-Saharan Africa [24]. In contrast, another study found that users, particularly women, thought modern contraceptives were less efficient in preventing pregnancy and were associated with concerns regarding infertility and cancer risks [22].

Contraceptive use exhibits substantial variation among different zones in Tanzania. The Southern and South West Highlands had higher rates of modern contraception use (38% and 32%), indicating better family planning awareness and access. In contrast, Zanzibar has the lowest prevalence rate at 8%, probably due to religious views that contraceptives opposing with religious beliefs and fears of infertility [23,25]. The low level of modern contraceptive use in Zanzibar may also be influenced by limited awareness regarding the utilization of male contraceptive methods and family planning [26]In addition, the role of culture, norms and gender power dynamics may have contributed to the variation in the contraceptive use between zones [27–29].

   

PLOS Global Public Health

**Table 3. Logistic regression on demographic and socio-economic factors on the use of contraceptive method among men aged 15-49 years in Tanzania (Using TDHS 2022).**

| Variable | | Contraceptive use among men | | | | | |
|---|---|---|---|---|---|---|---|
| | | OR | 95% CI | P-value | aOR | 95% CI | p-value |
| Age | 15–19 | Ref | | | Ref | | |
| | 20-24 | 4.06 | 2.97-5.55 | 0.001 | 2.97 | 2.19-4.04 | <0.001 |
| | 25-29 | 4.20 | 3.14-5.62 | 0.001 | 2.96 | 2.11-4.15 | <0.001 |
| | 30-34 | 4.47 | 3.19-6.27 | 0.001 | 3.11 | 2.11-4.61 | <0.001 |
| | 35-39 | 4.24 | 3.05-5.90 | 0.001 | 3.12 | 2.07-4.71 | <0.001 |
| | 40-44 | 3.74 | 2.68-5.23 | 0.001 | 2.89 | 1.84-4.53 | <0.001 |
| | 45-49 | 4.38 | 3.04-6.32 | 0.001 | 3.08 | 1.90-5.01 | <0.001 |
| Residence | Urban | Ref | | | Ref | | |
| | Rural | 0.77 | 0.64-0.93 | 0.006 | 0.84 | 0.67-1.07 | 0.156 |
| Education | None | Ref | | | Ref | | |
| | Primary | 2.37 | 1.81-3.11 | 0.001 | 1.88 | 1.41-2.51 | <0.001 |
| | Secondary | 2.40 | 1.79-3.23 | 0.001 | 2.04 | 1.46-2.84 | <0.001 |
| | Higher | 5.27 | 3.33-8.35 | 0.001 | 2.94 | 1.79-4.84 | <0.001 |
| Wealth | Poor | Ref | | | Ref | | |
| | Middle | 1.41 | 1.14-1.75 | 0.002 | 1.23 | 0.98-1.56 | 0.047 |
| | Rich | 1.78 | 1.46-2.18 | 0.001 | 1.42 | 0.92-1.46 | 0.012 |
| Occupation | Unemployed | Ref | | | Ref | | |
| | Employed | 3.92 | 2.61-5.89 | 0.001 | 2.01 | 1.34-3.02 | <0.001 |
| Marital status | Not in union | Ref | | | Ref | | |
| | In union | 1.38 | 1.17-1.63 | 0.001 | 0.60 | 0.27-1.32 | 0.21 |
| Parity | 0 | Ref | | | Ref | | |
| | 1-2 | 2.10 | 1.78-2.49 | 0.001 | 1.51 | 1.12-2.02 | 0.006 |
| | 3-4 | 1.89 | 1.51-2.35 | 0.001 | 1.53 | 1.05-2.24 | 0.029 |
| | 5 and above | 1.85 | 1.49-2.31 | 0.001 | 1.62 | 1.08-2.43 | 0.021 |
| Desire for another child | Have another | Ref | | | Ref | | |
| | Undecided | 1.09 | 0.75-1.57 | 0.658 | 1.10 | 0.73-1.65 | 0.629 |
| | Wants no more | 1.48 | 1.14- 1.93 | 0.003 | 1.40 | 1.05-1.88 | 0.022 |
| | Sterilized | 0.78 | 0.66-0.932 | 0.006 | 1.08 | 0.58-1.88 | 0.650 |
| Knowledge on contraceptive methods | No | Ref | | | Ref | | |
| | Yes | 3.81 | 2.88-5.04 | 0.001 | 1.95 | 1.45-2.62 | <0.001 |
| Heard family planning on radio | No | Ref | | | | | |
| | Yes | 2.07 | 1.77-2.42 | 0.001 | 1.39 | 1.16-1.66 | <0.001 |
| Contraception is women's business | Disagree | Ref | | | Ref | | |
| | Agree | 1.06 | 0.90-1.25 | 0.475 | 1.11 | 0.92-1.34 | 0.274 |
| | Don't know | 0.28 | 0.98-0.40 | 0.001 | 0.82 | 0.53-1.26 | 0.356 |
| Women using contraception become promiscuous | Disagree | Ref | | | Ref | | |
| | Agree | 0.98 | 0.84-1.15 | 0.839 | 0.95 | 0.79-1.14 | 0.575 |
| | Don't know | 0.18 | 0.12-0.27 | 0.001 | 0.42 | 0.27-0.66 | <0.001 |
| Frequency of listening radio | Not at all | Ref | | | Ref | | |
| | Less than once a week | 1.48 | 1.21-1.80 | 0.001 | 0.98 | 0.77-1.23 | 0.845 |
| | At least once a week | 1.57 | 1.29-1.90 | 0.001 | 0.85 | 0.68-1.06 | 0.140 |
| Frequency watching television | Not at all | Ref | | | Ref | | |
| | Less than once a week | 1.53 | 1.23 | 0.001 | 1.11 | 0.88-1.41 | 0.373 |
| | At least once a week | 1.70 | 1.38 | 0.001 | 1.16 | 0.92-1.46 | 0.222 |

*(Continued)*

**Table 3.** (Continued)

| Variable | | Contraceptive use among men | | | | | |
|---|---|---|---|---|---|---|---|
| | | OR | 95% CI | P-value | a0R | 95% CI | p-value |
| Discussion with health worker about contraception | No | Ref | | | Ref | | |
| | Yes | 1.41 | 1.14-1.73 | 0.001 | 1.01 | 0.81-1.26 | 0.953 |

Furthermore, it has been demonstrated that both age and educational progress influence contraceptive utilization, suggesting that maturity and educational attainment may promote family planning among Tanzanian men a pattern consistent with findings from other settings [30]. This emphasized the importance of education as a key element in improving the adoption of contraceptives and highlights the necessity for reproductive health education among adolescents to prevent unintended pregnancies [17]. The findings showed that as education increases also the use of family planning increases. This suggest that in order to improve the use of family planning among Tanzanian men increasing education is key. This finding is similar to the findings reported elsewhere [17,30]. Moreover, with advancing age, it becomes increasingly probable that a man has initiated a family and has achieved the desired number of children compared to younger individuals. Furthermore, men from moderate to affluent socioeconomic backgrounds and those who are employed demonstrate a greater inclination towards using modern contraceptives in contrast to those from less privileged backgrounds. Increased wealth and employment enhance an individual's ability to afford modern family planning methods compared to those who are poorer or unemployed as it is supported by another study in Ghana [1]. Additionally, the majority of employed and affluent individuals are more likely to have received education, good income and exposure to family planning awareness, thereby increasing their likelihood of encountering family planning promotions [31]. This is in contrary to the study done in Ethiopia [32] where, the wealth index did not show an independent association with the use of modern contraceptives.

Moreover, having a higher number of children and expressing a desire to cease having more children correlates with increased contraceptive utilization compared to those with fewer children or no child and no such desire, a trend also observed in other studies [2,30]. This finding aligns with another study, which found that visiting family planning facilities increased the likelihood of contraceptive use among men [2]. Interestingly, men's perception that women who use contraception are promiscuous influences their own contraceptive behavior, with those who do not hold such perceptions being less likely to utilize contraceptives. However, the study findings contradicted with the findings reported 60% contraceptive use among men in Kibaha, Tanzania [4]. Though the time of the study may have led to this remarkable difference in the prevalence as time differences among the two studies time is six years. Furthermore, the difference in coverage lie in the fact that the previous study in Kibaha focused on only one semi-urban district, whereas the current study covered the entire country, including rural, semi-urban and urban settings. Additionally, the two studies used different study populations; the Kibaha study involved only married men, whereas the current study involved the general male population across the country.

The study taps its uniqueness by assessing the use of contraceptive among men who are culturally taken to be the main decision makers and resources suppliers in the sexual relationship and family. This also accounts for all issues related to healthcare utilization, including reproductive health issues particularly decisions about whether or not to use contraceptives. This study took advantage of the gap left by many other studies that assess FP use based on women who are often reported to be merely implementers of decisions already taken by men [2,7,8,10,33].

The observed findings align with the framework used (Fig 1), highlighting the role of age, education level, wealth quantile, occupation, parity, exposure to FP via radio and knowledge on contraceptive methods in increasing contraceptive use among men. In addition, as per framework, rural residence showed to be a hindrance to contraceptive use among Tanzania men. Therefore, the alignment of findings with the framework, provides a comprehensive understanding of the factors linked to contraceptive use among men in Tanzania.

**Strengths and limitations of the study**

The cross-sectional nature of the study limited the establishment of causal-effect relationships between the variables.

The study analyzed the secondary data hence could inherently take the errors that are attached to the sampling design and other biases committed during data collection. Yet, overall, the use of national survey data as well as the use of weighting offers an advantage of eliminating many biases typically associated with pooling observational data, selection, and measurement bias.

The secondary nature of the study limited the analysis only to variables that were collected during the survey. Therefore, some of the variables could not be included due to not being included in the survey though acknowledged to be linked with use of contraceptives among men.

However, the study draws its strengths in the use of large DHS dataset which enabled authors to determine national representative estimates. In addition, the use of data collected through a standardized questionnaire applied in over 90 low and middle income countries (LMICs) worldwide enhances comparability and reliability of the findings reported in the current study.

## Conclusion

The use of contraceptive among Tanzanian men is generally low 26% and its use was mostly associated with the social demographic factors of education level, type of place of residence, number of children, employment, and the wealth as well as hearing FP messages on the radio.

**Policy implications of the study findings**

Based on the findings highlighted in the study, the government of Tanzania through the ministry of Health in collaboration with other stakeholders should establish tailored programs targeting men to increase their education level particularly health education focusing comprehensive reproductive health including the use contraceptive. Men should individually start to be concerned and participate in the reproductive health services like attending ANC and Postnatal care with their partners, whereby during the care, contraceptive use takes major part.

Future studies are recommended to provide detailed information on the specific types of contraceptives methods used by men in Tanzania. Additionally, qualitative studies should explore the underlying issues that prevent men from using contraceptives.

## Acknowledgments

The authors appreciate the custodian of DHS data for allowing the analysis that resulted in the completion of the current study.

## Author contributions

**Conceptualization:** Pankras Luoga, Jovinary Adam, Siri A Abihudi.

**Data curation:** Jovinary Adam.

**Formal analysis:** Jovinary Adam.

**Methodology:** Pankras Luoga, Jovinary Adam.

**Writing – original draft:** Pankras Luoga, Jovinary Adam, Siri A Abihudi.

**Writing – review & editing:** Pankras Luoga, Siri A Abihudi.

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
