## [Decision Letter · Decision Letter 0]

4 Feb 2025

PGPH-D-24-02612

PREVALENCE AND DETERMINANTS OF CONTRACEPTIVE USE AMONG MEN IN TANZANIA: ANALYSIS OF TANZANIA DEMOGRAPHIC AND HEALTH SURVEY 2022

Dear Dr. Luoga,

Thank you for submitting your manuscript to PLOS Global Public Health. After careful consideration, we feel that it has merit but does not fully meet PLOS Global Public Health’s publication criteria as it currently stands. Therefore, we invite you to submit a revised version of the manuscript that addresses the points raised during the review process.

Reflect on each of the reviewers' comments and ensure that you address them all. Please submit your revised manuscript by Mar 21 2025 11:59PM. If you will need more time than this to complete your revisions, please reply to this message or contact the journal office at globalpubhealth@plos.org. Please include the following items when submitting your revised manuscript:

We look forward to receiving your revised manuscript.

Kind regards,

Adriana Biney

Academic Editor

Journal Requirements:

1. We have amended your Competing Interest statement to comply with journal style. We kindly ask that you double check the statement and let us know if anything is incorrect. 

If you did not receive any funding for this study, please simply state: “The authors received no specific funding for this work.”"

3. In the online submission form, you indicated that "The data used in the analysis are available online and can be requested from the DHS custodian”. 

3. Uploaded as supplementary information.

4. Please insert an Ethics Statement at the beginning of your Methods section, under a subheading 'Ethics Statement'.

5. Please upload a copy of Figures 1 and 2 which you refer to in your manuscript. Or, if the figure is no longer to be included as part of the submission please remove all reference to it within the text.

Please provide separate figure files in .tif or .eps format.

Additional Editor Comments (if provided):

Reviewers' comments:

Reviewer's Responses to Questions

**Comments to the Author**

1. Does this manuscript meet PLOS Global Public Health’s publication criteria ? Is the manuscript technically sound, and do the data support the conclusions? The manuscript must describe methodologically and ethically rigorous research with conclusions that are appropriately drawn based on the data presented.

Reviewer #1: Yes

Reviewer #2: Partly

2. Has the statistical analysis been performed appropriately and rigorously?

Reviewer #1: Yes

Reviewer #2: No

3. Have the authors made all data underlying the findings in their manuscript fully available (please refer to the Data Availability Statement at the start of the manuscript PDF file)?

Reviewer #1: Yes

Reviewer #2: Yes

4. Is the manuscript presented in an intelligible fashion and written in standard English?

Reviewer #1: No

Reviewer #2: No

5. Review Comments to the Author

Reviewer #1: REVIEW OUTLINE FOR THE MANUSCRIPT: PREVALENCE AND DETERMINANTS OF CONTRACEPTIVE USE AMONG MEN IN TANZANIA: ANALYSIS OF TANZANIA DEMOGRAPHIC AND HEALTH SURVEY 2022.

JOURNAL: PLOS GLOBAL HEALTH.

Manuscript Number: PGPH-D-24-02612.

1. Summary of the research and your overall impression

This study is impressive and it plays a significant role of men involvement in reproductive health in a culturally sensitive societies like Tanzania where reproductive health is more termed and perceived as a women thing and not a men thing. Therefore, this study is a catalyst that portrays the essences of men involvement in reproductive health for the health and well being of a family and a strong economic nation and if the proposed study interventions are taken into consideration by the Tanzanian government which consists of more youthful population, we are highly likely to have less childhood pregnancies' and strong families.

2. Discussion of specific areas for improvement

2.1Title

The Title clearly explain the study topic and scope.

2.2 Abstract

The abstract is well structured with a good format flow of Background, Objectives, Methods, Results and Conclusions. However, at the background section, the authors have well elaborated the global positive implications of contraceptives with more focus on women excluding men who are crucial and of great interest for this study. Hence, the author should consider adding a brief information/explanations of the rationale of contraceptives to men for health and well being and especially to a cultural sensitive settings like Tanzania.

The objective is clearly elaborated and the authors have tried to structure the method section by considering study type, design and data collection methods, variables and data analysis techniques. However, the authors should consider expanding briefly the data analysis techniques in line with the study objective, the author should consider a brief explanation of how prevalence was accessed then determinants of contraceptives and consider mentioning the Chi-square test and logistic regression at the abstract because there are the main and crucial test used in this study analysis to access the associations.

Therefore, the authors could consider restructuring and a little bit expanding the methods part in the abstract and here is an example of a suggestion: "This is a cross-sectional study design that deployed secondary study approach by obtaining data collected through TDHS 2022 where a total of 5763 men were obtained......... In this study, a dependent variable was contraceptive use and independent variables were men's demographic and socio-economic characteristics........ was used to examine the prevalence of contraceptive use among men while chi-square test examined the association of.... and logistic regression examined.... and p-value <0.05 determined a significant factor".

The results and conclusions clearly and are briefly well elaborated.

2.3 Introduction

The authors should consider the following major and minor recommendations at the introduction section: Firstly, in reference to the article used to develop the idea in the first sentence “The economic prosperity, well-being of individuals, and environmental sustainability of a nation are contingent upon its capacity to effectively manage population growth”, the author should consider modifying the sentence by considering adding the trend of population growth in Africa/Sub Saharan Africa and how contraceptives can play a positive role in the economic prosperity, wellbeing of individuals and environmental sustainability.

For clarity reasons, the authors should consider restructuring the sentence “This has led to the promotion of diverse contraceptive methods in Sub-Saharan Africa, where individuals used to have a high number of children in the past, regardless of their gender ”, referring to the references used to develop the idea, the authors should consider restructuring this sentence by first explaining what traditional and modern contraceptive imply in Sub-Saharan Africa settings and how it has transformed over time because in the results and in the discussion part, the authors have argued about the traditional and modern contraceptives means while here in the background it is not addressed at all. For better understanding, the authors should consider re-reading the reference article no.17, a study conducted in Philippines to see how traditional and modern contraceptives are explained.

Besides, for the sake of clarity, the authors should consider a brief clarification of the example "where individuals used to have a high number of children in the past, regardless of their gender" by adding it up compared to today or? and if it is compared to today that people prefer few children, is it the same in rural and urban areas, given that most of our study subjects live in rural areas, and has contraception played a role in people preferring few children now compared to the past, or are there other factors that facilitate this? The authors should consider adding up a brief explanations that mention specific contraceptive methods commonly available, and mostly used in Tanzania setting before explaining the reasons for the rise in contraceptive uses in Tanzania/ before the sentence “Recently, there has been a rise in the variety of contraceptive methods available in Tanzania, and the desire to use the methods differs across different age groups, sex and geographical location ”.

For cohesion, coherence and duplication reasons, the authors should consider deleting the sentence “This study aims to investigate the factors what is behind the limited access of contraceptives for male in Tanzania”.

2.4 Methods

Overall the statistical analysis methods chosen are appropriate to examine the research questions. However, the authors need to clarify some few major and minor issues as follows: The authors should consider adding time intervals for implementing TDHS after the sentence “In Tanzania, the program is implemented by the National Bureau of Statistics (NBS) with the financial support from the United States Agency for International Development (USAID)” because that provide a picture and backup arguments on the trend of contraceptives use over time when compared to other studies that explored the same thing (contraceptive use among men) in Tanzania using the TDHS data.

The authors should consider adding explanations for the attributes/characteristics considered for the selected regions, as this may be related to why contraceptive use in Tanzania is geographically uneven.

The authors should consider adding an explanation of what criteria/attributes/characteristics that were considered in selecting the household, as this may be related to factors to determining contraceptive use and study limitations.

For clarity, the authors should consider adding how long the TDHS data collection took place before the paragraph “Interviews were conducted with all women and men between the ages of 15 and 49 who were usual residents or visitors slept in the selected households the night preceding the day of the survey……… Then the authors should consider specifying what types of interviews were deployed in the TDHS, whether they were structured, unstructured or semi-structured interviews, as to some point this could determine the generalizability and validity of the data, as this information is available in the TDHS website.

For structure and format reasons, the authors should consider shifting this explanations “No additional ethical approval was required beside the ethical considerations followed by DHS surveys. Ethical approval was obtained from participants prior to data collection by DHS program. The confidentiality and privacy of respondents are rigorously upheld throughout the DHS survey. Concerning this study; the MEASURE DHS approved the use of the datasets after reviewing our concept note that was submitted to them” to the ethical sections.

The authors have clearly explained the availability of the data used for this study by providing an accessible link and mentioning that "the datasets are available to the public free of charge from the DHS programme ("https://dhsprogram.com/data/new-user-registration.cfm)". However, the authors did not clearly state the procedures used to obtain the data, as in the ethical issues section the authors state that "permission to use the data was requested from the DHS custodian USAID MEASURES" implying that there were procedures involved. Therefore, for reasons of reliability, the authors should consider mentioning the procedures used to acquire the data.

Based on this sentence “In addition, this analysis involved some variables that were used in other similar studies conducted previously ”. For clarity reasons, the authors should further explain why they considered the variables mentioned for this study analysis, is it just because similar studies used the same variables? Why authors considered one study from Tanzania and the other from Philippines in selecting the variables for analysis?

For reasons of coherence, consistency and reliability, authors should consider moving the 'conceptual framework' to after the explanations of variables in the methods section. This should be followed by explanations of study bias. The authors should consider explaining any bias identified in this study and how the bias was addressed, because in the limitations sections the authors mentioned few biases in this study, but also, according to the STROBE checklist for cross-sectional studies, the authors should report any bias identified and explain how they addressed the bias.

Then there is the statistical analysis section. In the statistical analysis section, and in line with the study objectives, the authors might consider first explaining what statistical methods were used to examine prevalence, as it is not mentioned before the explanations of Chi-square and logistic regressions that were used to assess the associations of numerous factors facilitating contraceptive use.

2.5 Results

The authors have organized the descriptive table well and the main findings comprehensively reflect the objectives of the study. However, it is unclear and not mentioned which criteria the authors of this study used to group the study subjects as poorest, middle and richest. The caption of figure 2: “Figure 2: Prevalence of contraceptive use among the methods "it is unclear with phrasing issues. Hence, should it be…. a suggestion “Prevalence of contraceptive use methods among men”.

2.6 Discussions

The discussion comprehensively summarizes the findings of the study and links the results to other studies. However, some minor changes may be required for clarity and external validity reasons, for example, in the limitation of the study category, the sentence "The study analyzed the secondary data and therefore could inherently take the errors that are attached to the sampling design and other biases committed during data collection". The authors should consider specifically mentioning the sampling errors and biases that were committed during the data collection of this study.

In addition, for improvement and external validity purposes, the authors should consider mentioning potential variables that are not included in the Tanzania Demographic Health Survey (TDHS) and other studies, but are of great importance in assessing contraceptive use among men.

2.7 Conclusion

The conclusion is consistent with the study aims and results. However, for reasons of comprehensiveness and validity, the authors should consider amending the conclusion section to include explanations of the variables that were not associated with contraceptive use among men in Tanzania.

2.8 Recommendations

For reasons of comprehensiveness and based on the results of this study, the authors should consider proposing strategies to the government and other stakeholders to reach out to men for the health education interventions focusing on comprehensive reproductive health including contraceptive use. The strategies could be either through community outreach, as most of the study participants lived in rural areas or through media campaigns such as radio, television.

2.9 Other information

2.9.1 Ethical issues

Authors should consider adding explanations of how the data were handled after they were obtained from the TDHS and how long the data obtained were kept, in accordance with the Tanzanian research ethics law.

3. Confidential comments for the editors

The authors should consider a language expert for more language quality.

Reviewer #2: Title: PREVALENCE AND DETERMINANTS OF CONTRACEPTIVE USE AMONG MEN IN TANZANIA: ANALYSIS OF TANZANIA DEMOGRAPHIC AND HEALTH SURVEY 2022

Journal: Plos One Global Health

Date: 26th January, 2025

Reviewers Comments

I. Introduction

• Rationale: The authors include all contraceptive methods (modern and traditional) without providing a clear justification. This choice is unconventional, as modern contraceptive methods are typically prioritized in public health research for their efficacy and relevance to policy. If intentional, the authors should explain why traditional methods are included. For instance, are they addressing a gap in understanding cultural preferences?! Provide a clear rationale for analyzing all contraceptive use

• Redundancy and Lack of Focus: Several points are repeated unnecessarily, such as the increase in contraceptive methods and variations by age or residence. This repetition should be streamlined. Remove redundant information and focus on factors directly related to men’s contraceptive use.

• The inclusion of DHS statistics on women’s contraceptive use to deviate from the main topic. The focus should be on men’s contraceptive behaviors to align with the study’s objectives.

• Language Issues: Sentences like “factors what is behind the limited access of contraceptives for male” are grammatically incorrect and unclear. Phrasing should be revised for professionalism and clarity.

• This section lacks references to support the statements made. For example: that claim that "society has witnessed an increase in types of family planning methods in Tanzania" ; "the use of family planning without physicians’ consultations may have undesirable negative effects on the users"

II. Methods

• The methods section should clearly specify which men were included in the study (e.g., all men aged 15–49, only sexually active men) and who was asked questions about contraceptive use. It is unclear whether non-sexually active men were excluded or categorized differently, which is critical for understanding the applicability of the findings.

• The dependent variable is broadly defined as "use of Family Planning method,", author needs to note the difference between Family planning and contraceptive use! but also, it does not specify whether this includes modern, traditional, or all methods. Clearly define the dependent variable. If the study includes all contraceptive methods, provide a rationale for this choice and clarify how methods were categorized.

• The study mentions that variables with missing data were treated as "missing," but it does not explain how missing data were handled in the analysis (e.g., exclusion, imputation)

• The conceptual framework is presented (Figure 1), but its development and relevance to the study are not explained in detail. It is unclear how the framework was derived or why specific factors (e.g., economic, demographic, and attitudinal factors) were included. Additionally, the conceptual framework is not adequately integrated into the discussion.

III. Statistical analysis

• The analysis does not mention whether it accounted for the complex survey design or applied sampling weights to ensure representativeness (e.g use of svy command). This omission is significant since DHS data typically involve stratified, clustered sampling.

• The study mentions collinearity testing but does not specify the cutoff value or whether any variables were excluded based on collinearity.

IV. Results & Tables

• Table 1 is overly dense, combining socio-demographic characteristics, prevalence, and p-values in a single table.

Recommendation:

o Divide Table 1 into two separate tables:

One for socio-demographic characteristics and frequencies.

Another for associations with contraceptive use, including prevalence and p-values.

• P-values and CIs in Tables 2 and 3, they are inconsistently formatted. Use a consistent format i.e. number of decimal places, across all tables

• The results do not indicate whether sampling weights or adjustments for survey design.

V. Discussion

• The discussion repeats findings unnecessarily, such as the high proportion (82%) of men with knowledge of contraceptive methods and the emphasis on education promoting contraceptive use. Streamline the discussion by consolidating similar points and avoiding repetitive statements. For instance, the impact of education could be discussed once comprehensively rather than mentioned repeatedly.

• Some claims lack adequate referencing (e.g., the role of radios or differences in contraceptive use over time).

• The discussion lacks a clear connection between the findings and actionable policy or programmatic recommendations. Translate key findings into specific recommendations, such as improving access to education, targeting rural populations, or using community leaders to address cultural attitudes.

• The discussion mentions contradictory findings from Kibaha, Tanzania, without exploring the reasons for these differences beyond time gaps between the studies.

VI. Limitations

• The discussion of biases associated with secondary data (e.g., recall bias, social desirability bias) is vague and lacks specifics on their potential impact on the findings.

• The point about the secondary nature of the data limiting the analysis is repeated unnecessarily

VII. Conclusion and Recommendation

• The recommendations are vague, failing to provide specific strategies or interventions to improve contraceptive use among men.

• Recommendation needs to be as per study findings e.g “Men should individually start to be concerned and participate in the reproductive health services like attending ANC and Postnatal care with their partners” is not part of the study findings.

VI. List of Abbreviations

• The section is empty

6. PLOS authors have the option to publish the peer review history of their article (what does this mean? ). If published, this will include your full peer review and any attached files.

**Do you want your identity to be public for this peer review?** For information about this choice, including consent withdrawal, please see our Privacy Policy .

Reviewer #1: **Yes: ** Debora Yesaya

Reviewer #2: No

---

## [Decision Letter · Decision Letter 1]

6 Jun 2025

PGPH-D-24-02612R1

PREVALENCE AND DETERMINANTS OF CONTRACEPTIVE USE AMONG MEN IN TANZANIA: ANALYSIS OF TANZANIA DEMOGRAPHIC AND HEALTH SURVEY 2022

Dear Dr. Luoga,

Thank you for submitting your manuscript to PLOS Global Public Health. After careful consideration, we feel that it has merit but does not fully meet PLOS Global Public Health’s publication criteria as it currently stands. Therefore, we invite you to submit a revised version of the manuscript that addresses the points raised during the review process. See the attached files for the reviewers' comments.

We look forward to receiving your revised manuscript.

Kind regards,

Adriana Biney

Academic Editor

Additional Editor Comments (if provided):

Reviewers' comments:

Reviewer's Responses to Questions

**Comments to the Author**

1. If the authors have adequately addressed your comments raised in a previous round of review and you feel that this manuscript is now acceptable for publication, you may indicate that here to bypass the “Comments to the Author” section, enter your conflict of interest statement in the “Confidential to Editor” section, and submit your "Accept" recommendation.

Reviewer #3: (No Response)

Reviewer #4: (No Response)

2. Does this manuscript meet PLOS Global Public Health’s publication criteria ? Is the manuscript technically sound, and do the data support the conclusions? The manuscript must describe methodologically and ethically rigorous research with conclusions that are appropriately drawn based on the data presented.

Reviewer #3: Yes

Reviewer #4: Yes

3. Has the statistical analysis been performed appropriately and rigorously?

Reviewer #3: Yes

Reviewer #4: Yes

4. Have the authors made all data underlying the findings in their manuscript fully available (please refer to the Data Availability Statement at the start of the manuscript PDF file)?

Reviewer #3: Yes

Reviewer #4: Yes

5. Is the manuscript presented in an intelligible fashion and written in standard English?

Reviewer #3: No

Reviewer #4: No

6. Review Comments to the Author

Reviewer #3: (No Response) - See attachment

Reviewer #4: This is a needed article. I have highlighted areas to be addressed to improve the quality of the work in the attachment.

7. PLOS authors have the option to publish the peer review history of their article (what does this mean? ). If published, this will include your full peer review and any attached files.

**Do you want your identity to be public for this peer review?** For information about this choice, including consent withdrawal, please see our Privacy Policy .

Reviewer #3: **Yes: ** CATHERINE BERNARD BUNGA

Reviewer #4: **Yes: ** Ola Farid Jahanpour

---

## [Decision Letter · Decision Letter 2]

4 Sep 2025

Prevalence and Determinants of Contraceptive Use among Men in Tanzania: Analysis of the 2022 Demographic and Health Survey

PGPH-D-24-02612R2

Dear mr Luoga,

We are pleased to inform you that your manuscript 'Prevalence and Determinants of Contraceptive Use among Men in Tanzania: Analysis of the 2022 Demographic and Health Survey' has been provisionally accepted for publication in PLOS Global Public Health.

Best regards,

Julia Robinson

Executive Editor

Reviewer #3:

Reviewer Comments (if any, and for reference):

Reviewer's Responses to Questions

**Comments to the Author**

1. If the authors have adequately addressed your comments raised in a previous round of review and you feel that this manuscript is now acceptable for publication, you may indicate that here to bypass the “Comments to the Author” section, enter your conflict of interest statement in the “Confidential to Editor” section, and submit your "Accept" recommendation.

Reviewer #3: All comments have been addressed

2. Does this manuscript meet PLOS Global Public Health’s publication criteria ? Is the manuscript technically sound, and do the data support the conclusions? The manuscript must describe methodologically and ethically rigorous research with conclusions that are appropriately drawn based on the data presented.

Reviewer #3: Yes

3. Has the statistical analysis been performed appropriately and rigorously?

Reviewer #3: Yes

4. Have the authors made all data underlying the findings in their manuscript fully available (please refer to the Data Availability Statement at the start of the manuscript PDF file)?

Reviewer #3: Yes

5. Is the manuscript presented in an intelligible fashion and written in standard English?

Reviewer #3: Yes

6. Review Comments to the Author

Reviewer #3: (No Response)

7. PLOS authors have the option to publish the peer review history of their article (what does this mean? ). If published, this will include your full peer review and any attached files.

**Do you want your identity to be public for this peer review?** For information about this choice, including consent withdrawal, please see our Privacy Policy .

Reviewer #3: **Yes: ** CATHERINE BERNARD BUNGA
